# Test Structure Design for Defect Detection during Active Thermal Cycling

**DOI:** 10.3390/s22197223

**Published:** 2022-09-23

**Authors:** Ciprian Florea, Dan Simon, Adrian Bojiță, Marius Purcar, Cristian Boianceanu, Vasile Țopa

**Affiliations:** 1Infineon Technologies Romania & Co. SCS, 020335 Bucharest, Romania; 2Electrotechnics and Measurements Department, Technical University of Cluj-Napoca, 400027 Cluj-Napoca, Romania

**Keywords:** active thermal cycling, DMOS power transistor, smart power IC, failure mechanisms, integrated sensors, metallization fatigue, test structure, electro-thermal, thermo-mechanical

## Abstract

Integrated power ICs acting as smart power switches for automotive or industrial applications are often subjected to active thermal cycling. Consequently, they undergo significant self-heating and are prone to various failure mechanisms related to the electro-thermo-mechanical phenomena that take place in the device metallization. In this article a test structure consisting of a lateral DMOS transistor equipped with several integrated sensors is proposed for metallization fatigue assessment. The design of the test structure is presented in detail, alongside with design considerations drawn from the literature and from simulation results. The testing procedure is then described, and experimental results are discussed. The experimental data provided by the integrated sensors correlated with the electro-thermal simulation results indicate the emergence of a failure mechanism and this is later confirmed by failure analysis. Conclusions are further drawn regarding the feasibility of using the proposed integrated sensors for monitoring defects in power ICs.

## 1. Introduction

The current developments in the BCD (bipolar-CMOS-DMOS) manufacturing processes of power semiconductors for the automotive industry allow for higher integration and higher transistor density due to minimum detail shrinkage. These technologies allow the monolithic integration of bipolar devices for fast and precise analog functions and sensors, CMOS for digital functions and power devices (DMOS) for delivering high currents to various loads. In this way a new class of power integrated circuits (ICs) was created, widely known as smart power ICs [1,2]. The ICs manufactured in new BCD technologies operate at higher power densities due to their significantly reduced areas. As a consequence, they suffer from more pronounced self-heating, and non-uniform temperature distributions which lead to the formation of hotspots with junction temperatures reaching up to 400 °C [3,4]. This in turn leads to more severe electro-thermal [5,6,7] and thermo-mechanical effects [8,9,10,11,12]. Electro-thermo-mechanical phenomena caused by the intensive self-heating are the main cause for the emergence and evolution of various failure mechanisms which limit the safe operating area (SOA) and reduce the lifetime of power ICs. This is especially the case for devices operating as switches in automotive or industrial applications, where high power transients are delivered in short pulses (hundreds of microseconds to millisecond range). In this case, from the thermal point of view, the device metallization plays an important role. A cross-section of a typical metallization found in smart power ICs is presented in Figure 1.

The metallization consists of a stack of several layers of thin metal interconnected with vias with a thick metal layer on top, see also [2]. In a DMOS transistor, the stacked thin metal layers interconnected with vias are used to carry the source-drain current and distribute it as uniformly as possible across the device area. In other parts of the chip, the thin metal stack is used to route signals in digital and analog blocks; therefore, it is called signal metallization and is abbreviated as SM. The top thick metal layer is used to transport the high currents from the chip pins to the device with the minimum electrical resistance (low voltage drops), and also helps the currents to be uniformly distributed across the device area. Therefore, this metal layer is called power metal and is abbreviated as PM. An extensive study regarding the influence of metallization on thermal performance of such devices carried out on different metallization schemes in [13] indicates the important roles of the PM and SM. The power metal acts as a heat sink, effectively absorbing heat for short time intervals. The SM is in the heat path which is generated at the Silicon surface, and which propagates towards the PM. The SM also acts as an additional thermal capacitance, significantly reducing the device self-heating during short pulses, thus increasing the device electro-thermal SOA. Any failure mechanisms emerging in these regions would drastically reduce the device lifetime and this is the premise for the investigation carried out in this paper. 

Some of the most frequently manifested failure mechanisms in power ICs metallization which are associated with fast thermal cycling are the inter-metal-dielectric (IMD) cracking [8,9,10] and the PM cracking and delamination from the underlying SM [11,12], failure mechanisms which will be detailed in the next two paragraphs. For a better understanding of these failure mechanisms, one can refer to Figure 2. The figure represents a schematic top view of a lateral DMOS (LDMOS) metallization (Figure 2a), a cross-section along the XX’ axis (Figure 2b) and a cross-section along the YY’ axis, respectively (Figure 2c). The drain and source PM plates are colored in dark orange. The underlying SM is colored in dark green, and the dielectric is colored in beige.

### 1.1. IMD Cracking Failure Mode

A cross-section alongside the XX’ axis (Figure 2b) reveals two adjacent lines belonging to the topmost thin metal layer (TTML) of SM, right underneath the bottom edge of the PM. One TTML line is connected to the drain plate through large vias called power vias (PV), while the TTML line that carries the source signal connects to the source plate on a different plane. In devices subjected to fast thermal cycling, high temperature gradients appear alongside TTML lines (alongside YY’ direction in Figure 2a). At high temperatures the metal suffers plastic deformation and thermo-migration occurs on the direction of the temperature gradients from the cold regions to the hotter ones in the center. This leads to a thickening of the TTML up to 10%, in the hottest regions, as it is shown in [8,9]. The thickened TTML lines along with the PM plate exert stress on the oxide. When the fracture strength of the oxide is exceeded, the dielectric between two thickened metal lines will crack, as shown in Figure 2b, leading to a short-circuit. In the case of a LDMOS device, this means drain–source short-circuit, and hence, critical failure.

### 1.2. Power Metal Delamination/Disruption

During fast thermal cycling the PM can also be subjected to an even more severe plastic deformation due its greater thickness [14] compared to the one of the underlying SM. However, the peak temperature in PM is significantly lower (in most cases not high enough for the metal to enter in the plastic deformation domain). In addition, the plastic mold compound in the packaging contributes to the heat dissipation and may introduce other thermo-mechanical failure modes. This is mainly the reason why delamination is less associated with on-chip metallization and rather more frequently found at the packaging level in die attach [15,16], mold compound [17], and bonds [18]. In some cases, however, the delamination of the on-chip PM is identified, although in most situations it occurs after a far larger number of cycles than IMD cracking. It is considered to be a second-order contributor to chip failure. Thermal ratcheting of metal films has been studied in [19,20,21] and it has been concluded that metal delamination from an underlying structure occurs as a result of the high interfacial stresses caused by the mismatch between the coefficients of thermal expansion (CTE) of the metal and the underlying structure. In the schematic drawing in Figure 2c which is a cross-section along the YY’ axis, it is shown how such a mechanism would manifest in a power DMOS metallization. As the number of cycles increases, the PM starts to deform from the center of the device in the opposite direction (transition from phase 1 to 3 in Figure 2c), thickness variation may occur, and voids and cracks may appear inside the PM plate, propagating horizontally along its interface with the underlying SM as observed in [11,12]. The latter defects (cracking and voiding) lead to a reduction in the contact area through which heat is being transferred from silicon to the thermal capacitance of the PM, which increases device self-heating, eventually leading to thermal runaway. Although the two failure mechanisms (i.e., IMD cracking and PM delamination) often overlap and influence each other, it is possible to force one or the other to be more predominant either by careful metallization layout design or by adequate choice of operating conditions as the IMD cracking is known to be mainly driven by the temperature gradient alongside TTML lines, see [8], while PM delamination is mainly driven by a peak temperature rise in the metal plate, see [19,21].

### 1.3. Objective of the Current Work

In this paper a lateral DMOS transistor equipped with several integrated sensors was designed and tested in order to detect IMD cracking and PM delamination in the early stages. Such devices could be used in non-critical applications provided that the devices can be switched to a safer operating condition once the manifestation of such failure mechanisms exceeds certain thresholds.

The paper is structured as follows: In Section 2 the test structure is described in detail. Layout design considerations are discussed regarding the DMOS device and the integrated sensors. In Section 3 the testing procedure is explained. First some preliminary measurements are presented, then the adequate testing conditions for active thermal cycling (ATC) are determined. In Section 4 the postprocessed experimental results and failure analysis results are discussed. In Section 5 conclusions are drawn on the entire work carried out in this study to assess to what degree the proposed test structure is suitable for metallization reliability assessment.

## 2. The Test Chip

### 2.1. The DMOS Transistor

The device used as test structure consists of a DMOS transistor equipped with several integrated sensors. The transistor is a typical lateral DMOS manufactured in a standard BCD technology, such as the one described in [2]. The design is based on the work carried out in [10] in the interest to preserve as much as possible the physical structure (the active area, the aspect ratio, the split-gate feature) as well as the pinout, current capabilities, package compatibility, test system compatibility, and testing conditions. Therefore, the device is a split-gate DMOS with an active area (AA) of approximately 0.15 mm^2^ and an aspect ratio X/Y = 5. The split gate allows for partial deactivation of the AA in the center, allowing for only two possible configurations which will be labeled as:100%AA (100% of the active area operational);90%AA (90% of the active area operational, 10% of DMOS deactivated in the center);

From the work carried out in [10] it was deducted that partial deactivation of 10%AA in the center increases the lifetime of the device by approximately one order of magnitude; therefore, it was desired to include the option to switch the device to a safer operating condition once the emergence of a failure mechanism poses a serious threat. The integrated sensors are described in the following paragraphs.

### 2.2. The Temperature Sensor

First, a temperature sensor is required for monitoring the instantaneous temperature in the center of the device during ATC. Sensors must be small, fast, precise, and their interference with the normal operation of the device must be minimal. Although PN-junction sensors, such as the ones presented in [3,4] seem to be suitable candidates, they come with certain drawbacks: first, they must be electrically isolated from the rest of the device which leads to the creation of a small gap in the active area, and secondly, the sensors need to be routed separately which requires local modifications in the bottom layers of metallization. Both requirements would affect the source potential distribution which is crucial for device performance. In addition, according to the literature, the aforementioned failure mechanisms are more likely to manifest in the topmost layers of metallization; therefore, the temperature in these layers is much more relevant compared to the actual junction temperature. In conclusion, in the interest of preserving a uniform source potential distribution and to assess the temperature in the area of interest, the optimal approach is to place a metal meander in the TTML, right above the center of the device. The resistance of the metal meander is temperature-dependent and will be measured in a four-wire connection as shown in Figure 3. This allows the monitoring of the TTML temperature throughout the whole testing process. Nevertheless, the junction temperature, if required, can be estimated from an electro-thermal simulation (see [13,22,23,24]) once the simulation setup is calibrated so that the simulated peak temperature in the TTML resembles the measured one.

### 2.3. The Mechanical Sensors

In order to detect the IMD cracking (presented in Figure 2b) the sensor needs to act as a metal barrier which temporarily prevents full propagation of the crack from drain to source or vice versa, as Figure 4a suggests.

The solution adopted in [25,26] was a metal meander or multiple such meanders between adjacent metal lines on different layers which extend all across the entire AA. This approach comes with major drawbacks: (i) a sensing structure extending across the entire area of the device increases the metal-to-oxide ratio, rendering the device metallization far more prone to IMD cracking than desired (i.e., less reliable from the beginning), and (ii) it makes it somewhat difficult to obtain information regarding the precise location of the defect. The approach proposed in this paper consists of placing some thin metal lines in TTML between adjacent metal lines carrying different signals (i.e., source and drain signals) and routing them in groups on a different metal layer in a comb-like structure as suggested in Figure 4b. This approach allows the designer to preserve as much as possible of the metallization reliability as the presence of the sensors is not required everywhere and can be limited exclusively to the areas where it is more likely for the IMD cracking to appear. In conclusion, the width, the length, the density, and the grouping of the thin metal lines used as IMD cracking sensors can be conveniently adjusted in order to detect the desired failure mechanism while preserving a reasonable amount of the device metallization reliability. In order to make an assessment regarding an adequate placement of the sensors, the results of the thermo-mechanical simulations presented in [10,27] were taken into consideration. The simulations were carried out on almost identical metallization structures and the simulated stress and strain distributions suggested that the maximum plastic strain accumulation takes place in TTML, right underneath the PM bottom edge (along the XX’ axis in Figure 2a). In addition, the work carried out in [8,9] showed that the IMD cracking is a temperature gradient-induced failure mechanism, and the temperature gradient in TTML is the root cause of device failure. Based on these results, a good starting point is to extend the IMD cracking sensors underneath both drain and source PM plates to such a length so that they would provide reasonable coverage of the zones where hotspots would appear. In order to be able to decide over the length and the placement of the groups of such sensors, electro-thermal simulations were employed. The simulations were performed using a 3D simulator which is described in detail in [22,23]. The geometry was extracted from the 2D layout of the test chip, and layer thicknesses, material properties, and models were provided by the manufacturer. Power pulses such as the ones used in the test conditions specified in [10] were applied, and the temperature in TTML was assessed. The temperature distribution was plotted for both cases (100%AA and 90%AA) in Figure 5. Superimposed on the temperature distribution are the sensor groups (notice the thin white lines). 

The length and the grouping of the sensors was chosen in such a manner so that each group would contain the high-risk areas each hotspot (temperatures that exceed 280 °C). It is possible to force the IMD cracking to be the predominant failure mechanism either by widening the TTML lines or by placing the sensor closer either to the drain or to the source line. The proposed detection method offers some flexibility regarding its implementation as one may select which type of short-circuit (be it either source–sensor or drain–sensor) is more suitable for a specific application. The solution adopted in this paper was to place the sensors at approximately mid-distance between source and drain TTML lines. This approach allows the detection of both drain–sensor cracks and source–sensor cracks with almost equal probability. Since that is the case, each group of sensors were connected to a high-impedance external sensing circuitry which allowed for both potentials (be it drain or source) to be detected without further damaging the metallization once the short circuit appeared. 

In order to detect PM delamination, an embodiment of [28] was implemented as a delamination sensor which consists of isolating one via and routing the other end separately to monitor its electrical resistance, as shown in Figure 6. 

As the delamination occurred and progressed, it was expected that the electrical resistance of the sensor would increase with the number of cycles. In parallel, the temperature was monitored on the aforementioned temperature sensor to see to what degree the increase in resistance with the number of cycles correlated with an increase in peak temperature with the number of cycles, in this way validating the concept of PM delamination detection on one hand, and the fact that PM delamination increases self-heating on the other hand. For the adequate placement of the delamination sensors electro-thermal simulations were used as well, with the exception that this time the temperature in PM was the one of interest because PM disruption is driven by its peak temperature. The temperature distributions in PM for both scenarios (100%AA and 90%AA, respectively) are presented in Figure 7.

The hottest spots appeared in the drain PM plates as a consequence of the way the underlying TTML was routed (the constraint was to maintain a source potential distribution as uniform as possible). However, the device layout can be designed in such a manner so that the signal lines corresponding to the hotspots are carrying the source signal in case it is more convenient, given a certain application, to measure the via resistance at lower potentials, while still preserving source potential uniformity and device on-resistance, R_DS(ON)_. Hence, this detection method also offers some flexibility regarding its implementation. One extra care in layout design, however, is to provide enough space between the sensor and its surrounding TTML line, see Figure 6b, so that no IMD cracking can take place between the two, rendering the sensor useless. The isolated vias were placed as close to the hotspots as permitted by the design rules provided by the chip manufacturer.

### 2.4. Test Chip Overview

Placing all sensors (temperature and mechanical) on the same device was problematic because: (i) the temperature and delamination sensors occupy the same spot in TTML (see Figure 3 and Figure 6b), (ii) too many modifications in the metallization might affect device performance, and (iii) limitations (such as the number of pins accessible at the same time, for example) were imposed by the external test system and the adopted packaging solution. The best solution to adopt was to place two identical DMOS transistors on the same chip. One transistor was equipped with the integrated temperature sensor and will be referred to from now on as the calibration structure, while the other one was equipped with the integrated mechanical sensors and will be referred to from now on as the test structure. The transistors were measured separately. The calibration structure was provided with force and sense terminals for the drain and source of the DMOS as well as force and sense terminals for the temperature sensor. It was used to adjust the pulse length and amplitude in order to obtain the desired temperatures in TTML. The calibration structure was also used to calibrate the electro-thermal simulation setups needed for the further investigation of various testing scenarios. The test structure was composed of the DMOS transistor equipped with the IMD crack sensors and PM delamination sensors. The latter was subjected to ATC under the agreed testing conditions. The mechanical sensors were monitored in order to assess the occurrence and evolution of the aforementioned failure mechanisms. 

A top view of the test chip is presented in Figure 8. The test chip was encapsulated in a ceramic CDIP24 package. The CDIP24 package was adopted instead of a plastic package for the following reasons: (i) the ceramic package can withstand high ambient temperatures, (ii) optical inspection of the chip is required after several periods of testing. The transistors were bonded in such a manner so that the calibration structure and the test structure were interchangeable any time by simply rotating the package in the testing slot by 180°. This allows the user more flexibility in testing and reduces the wiring in the test system.

## 3. Testing Procedure

### 3.1. Preliminary Measurements

Preliminary measurements and sanity checks were required in order to ensure that all test chips functioned properly and to assess their parameter spread. The parameter spread was very important for our study because the test structures containing mechanical sensors were not equipped with sense terminals for drain and source, or with temperature sensors. Therefore, the calibration structure was used to determine the parameter spread. It is important to make sure that all devices exhibit virtually identical behavior for small variations in device parameters or testing conditions (variation of bond wire resistance, variations of supply voltages, variations of ambient temperature, etc.) without having to perform thorough time-consuming characterization on all devices. Small parameter spread is desirable as it makes the electro-thermal simulation setup calibration easier and offers more confidence in simulation results. 

The gate-to-source voltage (V_GS_) distribution for constant drain current (I_D_), constant drain-to-source voltage (V_DS_), and constant ambient temperature is presented in Figure 9a. One can observe that the spread was rather small, slightly over 1%. This result, in conjunction with the normed transfer characteristic presented in Figure 9b is necessary for electro-thermal simulation setup calibration and enables decisions over the preferred operating condition of the device, with respect to temperature compensation point (TCP) to be made [23]. TCP is the V_GS_ value for which I_D_ is independent on temperature.

Further, in Figure 10 the spread of I_D_ for constant V_GS_ and constant V_DS_, is presented for: 100%AA, 90%AA, and the current ratio defined as I_D-90%AA_/I_D-100%AA_. Reduced I_D_ spread ensured virtually identical power dissipation on all devices under the same testing conditions, while reduced spread in the current ratio provided reasonable confidence that when de-activating the central gate, the remaining active area was virtually identical on all devices.

The average temperature characteristics of the temperature sensor and the delamination sensor are shown in Figure 11. The temperature dependency of the electrical resistance was measured in the 20 °C ÷ 180 °C range and was in good agreement with the linear variation
R(T) = R_0_ × [1 + α × (T − T_0_)],(1)
where R_0_ is the resistance at T_0_ = 300 K and α is the temperature coefficient. The temperature characteristic is then extrapolated at the temperatures of interest, which are much higher, exceeding 400 °C. The temperature dependency of the electrical resistance of the delamination sensor was close to that of the temperature sensor, which suggests that the delamination sensor could also make a good candidate for measuring the temperature rise in PM, if provided with sensing pads.

The distributions of R_0_ for the temperature sensors and delamination sensors is presented in Figure 12. Again, the resistance spread was very small, less than 3%, which suggests that time-consuming characterization on each sensor can be avoided and it is reasonable to presume that all sensors virtually delivered the same output.

Some of the measurements were carried out directly on wafer and some were performed on packaged chips. No significant differences were noticed between the same characteristics measured in both cases, which is exactly the desired result. It is then a reasonable assumption that the ATC is the only contributor to device self-heating, and no contribution whatsoever is given by chip packaging or ambient temperature variation. This also makes it easier to model the thermal behavior of the device and its metallization. The calibration of the electro-thermal simulation setup is discussed in detail in [24] so it will not be approached here.

### 3.2. The Power Cycling Procedure

As testing in real operating conditions is time-consuming (virtually impossible), the test chips were subjected to fast thermal cycling in more aggressive conditions than those found in normal operation (e.g., higher ambient and higher junction temperatures) in the interest of accelerating the degradation process of the metallization and obtaining the desired information in a fairly reasonable amount of testing time. The measurement system used for stressing the devices under ATC was based on the one described in detail in [10] so it will not be extensively discussed in this paper. It roughly consists of an oven which sets the ambient temperature for the devices under test (DUTs), lab instruments such as: arbitrary wave generator (AWG), digital sampling oscilloscope (DSO), power supplies, multimeters, a gate driver for repeatedly switching the DUTs on and off, a microcontroller, and a computer. 

During each test, a lot consisting of eight chips, one calibration structure, and seven test structures were introduced in the oven. The oven temperature was set at a constant value. The thermal cycling system was started and the temperature sensor on the calibration structure was monitored until the on-chip temperature measured in the off-state reached a steady-state value (usually in a few thousands of cycles). Fine adjustments were then made to the supply voltage so that the power monitored on the calibration structure was settled to the desired value. The power and temperature were monitored on the calibration structure and were assumed to be the same for the test structures, given the low parameter spread discussed in Section 3.1. The IMD cracking and PM delamination sensors were monitored on the test structures. The selection of the adequate power pulse magnitude and ambient temperature was determined by trial. In Table 1, the three sets of testing conditions for the samples labeled from 1 up to 7 in Figure 13 are are listed.

The delamination sensor response is shown in the graph in Figure 13 which depicts the increase in the resistance with the number of cycles with respect to its initial value, as in
R_increase_ = (R_current_ − R_initial_)/R_initial_ × 100 (2)

In the beginning, the test conditions were those specified in the first set, Test 1 in Table 1, identical to those described in [10]. For the first 18 million cycles no IMD cracking and no delamination were observed whatsoever; therefore, the conditions were switched to those specified in the second set, Test 2 in Table 1. After 500 thousand cycles, no IMD cracking was observed; however, the delamination was substantial, a steep increase in the sensor resistance can be observed in the graph, and another additional mechanical phenomenon was observed on some samples in the form of crack formation and propagation in the PM plates (PM disruption) and some devices either failed critically (although it was not clear whether it was due to thermal runaway or source–drain short-circuit), either their delamination sensors interrupted (samples 1, 3, and 6). In addition, the temperature measured by the sensor was quite high, 465 °C, so the probability of slipping into thermal runaway was very likely. 

The second set of testing conditions presented multiple results with multiple possible causes which makes it hard to decide the predominant cause which led to the critical failure of the device. Therefore, with the devices that survived the second test (samples 2, 4, 5, and 7), the tests were continued with somewhat less severe testing conditions, those specified in the Test 3 set of Table 1. No undesired cracks appeared anymore, the temperature posed no risk of thermal runaway, and no steep increases in the sensor resistance were observed; hence, these conditions were considered to be suitable for testing and were adopted for the rest of the entire testing process.

## 4. Experimental Results

### 4.1. Delamination Sensor Resistance Variation and Self-Heating Increase

A total of 16 samples were subjected to several batches of 5 million cycles in 100%AA configuration. The reason for such an approach was the need for optical microscopy inspection and photographing of the device metallization after each set of 5 million cycles. In addition, it was desired to have several sets of at least two or three devices tested for the same number of cycles in different phases of aging available for failure analysis. This is the reason why not all devices were tested for the same amount of time. None of the chips failed during testing. However, the optical microscopy photographs presented in Figure 14 indicate a significant buckling of the PM plates and fracturing of the passivation layer which started to take place for all chips somewhere in the 10÷15 million cycles range.

The PM delamination is suggested by the increase in the delamination sensor electrical resistance, calculated as in (2), and plotted on the graph in Figure 15. The samples are labeled 8 to 23.

The resistance increase ranged from 20% and up to 70% with respect to its initial value. The temperature was monitored by the temperature sensor situated on the calibration structure. The self-heating of the device is defined as the temperature difference with respect to its steady-state value, as in
ΔT = T_sensor_ − T_steady-state_(3)

A close-up of the ΔT at the end of the power pulse is plotted in Figure 16, for the calibration structure.

It can be observed that the self-heating increased with the increasing number of cycles, which also suggests that PM delamination was taking place and had an impact on device self-heating. The self-heating tended to saturate after 15 million cycles, however. Moreover, 9 out of 16 samples exhibited a 10% increase in the resistance of the delamination sensor before or starting with 15 million cycles. This result suggests that one could adopt as a possible failing criterion either an increase in temperature by 10 °C or an increase in resistance by 10%. After this threshold is exceeded, the device ought to be switched to 90%AA to prolong safe operation. Tests were performed as well on chips with 90% active area, but since the delamination sensors on the lateral hotspots were not routed, they could not be measured. Hence, the only assessment that could be made was the comparison of the optical microscopy photographs, simulation plots, and failure analysis results. A comparison between optical microscopy images depicting PM plates subjected to ATC for devices operating at 100%AA and 90%AA, respectively, along with the according simulated temperature distributions, is shown in Figure 17. 

It can be noticed that the devices operating at 100%AA exhibited a far more pronounced PM plate buckling than the ones operating at 90%AA. This result correlates very well with the simulated temperature distribution in PM. It can be observed that the peak temperature in PM for 90%AA was approximately 15 °C smaller with regards to that in 100%AA case and the temperature distribution was somewhat more uniform in the 90%AA case. This result confirms once again that the main driving factor for PM delamination is the peak temperature in PM. 

### 4.2. Failure Analysis Results

Three samples were sent to failure analysis for Focused Ion Beam (FIB) cross-sectioning and Scanning Electron Microscopy (SEM) inspection of the metallization: (i) one unstressed sample used as reference structure for comparison, (ii) one sample stressed for 30 million cycles in 100%AA operation, and (iii) one sample stressed for 30 million cycles in 90%AA operation. The samples were cross-sectioned parallel to the YY’ axis in Figure 2a in such a manner so that the cut passed right through the center of the delamination sensors. The unstressed sample and the sample operated at 100%AA were cut right through the center and the sample operated at 90%AA was cut through one of the lateral sensors situated in one of the hotspots presented in Figure 17. 

The cross-sections depicting the PM, the delamination sensor and the TTML are presented in Figure 18.

In Figure 18a the normal structure of an unstressed sample is shown. The PM plate was connected to the TTML through the PV which was the proposed delamination sensor. On top of the PM plate there were two more layers: a coating layer and the passivation layer on top. The stressed samples presented in Figure 18b,c show severe damaging compared to the unstressed structure: (i) the passivation layer is fractured in both cases, (ii) the polycrystalline-like structure of the PM is severely disrupted, (iii) pronounced voids and cracks have appeared inside PM and at its interface with the underlying structure. Delamination (e.g., the interfacial crack) of the PM is visible in both cases, although it seems more evident in the 90%AA operation as the close-up around the sensor in Figure 19 suggests. 

One might argue that such a result is rather counter-intuitive since the 100%AA operation is supposed to be more stressful for the metallization. In reality, in Figure 18b it can be seen that in the case of 100%AA operation the coating material suffered a severe lift-off. In addition to that, the passivation layer was fractured, the CDIP package was not hermetically sealed, therefore, severe oxidation took place in PM, overlapping on the original failure mechanism. It can be seen by comparing Figure 18b,c, that in the case of 90%AA operation, the coating material did not lift off and the PM retained more of its original polycrystalline-like structure. On the other hand, in the case of 100%AA operation a “bird’s beak” appeared inside PM and the metal did not retain its polycrystalline-like structure. The close-up in Figure 19a also shows signs of oxidation of PM and PV. However, although it is a more complex mechanism, oxidation adds to the same problem—it hinders heat transfer in the thermal capacitance of the PM and increases the PV resistance. The cross-sectioned sample operating at 100%AA is the sample labeled “17” in the graph in Figure 15. The coating layer liftoff followed by oxidation explains very well the increase in the electrical resistance by approximately 60%, which indicates that delamination and other additional failure mechanisms (such as oxidation) that might affect PM can be monitored with the proposed sensing method.

Unfortunately, no short-circuit caused by IMD cracking could be detected electrically and none of the tested devices failed, as mentioned before. In Figure 20 is presented a cut along the TTML line starting from the center of the device (the hotspot) and continuing deep underneath the opposite (source) PM plate, for a longer distance than the length of the thin metal lines used as IMD cracking sensors, as the layout in Figure 20d suggests.

The TTML lines under the source PM plate are relevant in this case because they are carrying drain signals; therefore, they are not connected with the source PM plate, which makes these TTML lines hotter, as the temperature distribution in Figure 5 suggests. It can be observed that for the stressed samples (Figure 20b,c) no visible change in thickness can be noticed, not even after 30 million cycles. As well, no spatial variation in thickness along the TTML line was observed, despite the high temperature gradients shown in Figure 5. It can be concluded that in these testing conditions it is very unlikely that any IMD cracking will be captured because the plastic strain accumulation in TTML is negligible, despite the high thermal stress.

## 5. Conclusions

In this paper a test chip consisting of a power DMOS device equipped with several integrated sensors was designed and tested in the interest of validating the use of integrated sensors for detecting two of the most frequent failure mechanisms related to fast thermal cycling: IMD cracking and PM delamination.

Layout design considerations drawn from the literature and backed by electro-thermal simulations were discussed and implemented in the design of the test chip.

Preliminary measurements (device and sensor characterization and sanity checks) were performed in order to ensure very-well controlled testing conditions and easy and accurate electro-thermal modelling of the device.

Electrical measurements performed during ATC indicated an increase in the delamination sensor resistance ranging from 20% up to 70% of its original value, which indicates that PM delamination is taking place and can be detected with the proposed sensing method.

Device self-heating increased by approximately 10 °C after ATC, which suggests that PM delamination leads to increased self-heating by hindering heat transfer in the thermal capacitance of the PM.

Finally, the failure analysis results confirmed that severe PM delamination was taking place, in this way validating the proposed detection method. Moreover, failure analysis confirmed that 90%AA is a less severe operating condition than 100%AA and that the main driving factor for PM delamination is the peak temperature in PM.

No IMD cracking was detected, as failure analysis confirmed no plastic deformation in TTML. It can be concluded that in this particular case, the manufacturing technology was simply too reliable to spot such a failure mechanism, the only observable failure mechanism being PM delamination.

## Figures and Tables

**Figure 1 sensors-22-07223-f001:**
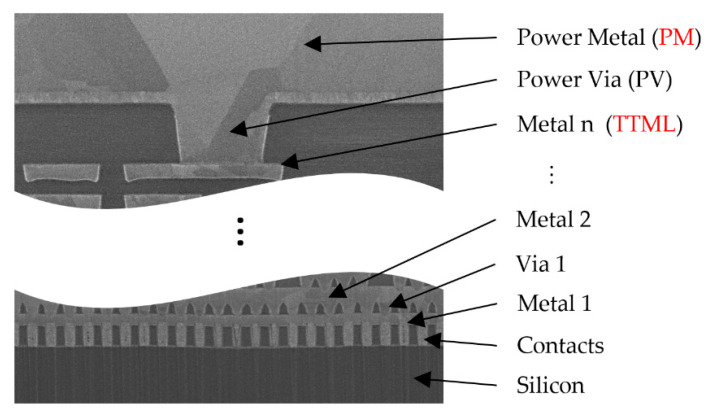
Cross-section through a typical metal stack of a smart power IC.

**Figure 2 sensors-22-07223-f002:**
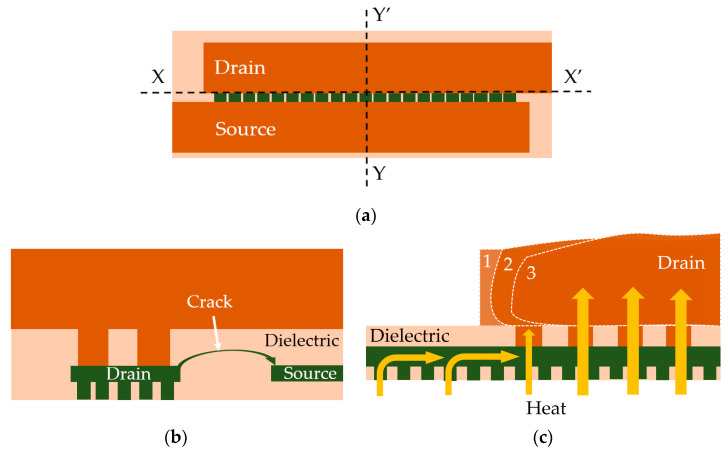
(**a**) Top view of the DMOS device metallization; (**b**) cross section alongside XX’ axis, illustrating the IMD cracking; (**c**) cross section alongside YY’ axis illustrating the PM delamination.

**Figure 3 sensors-22-07223-f003:**
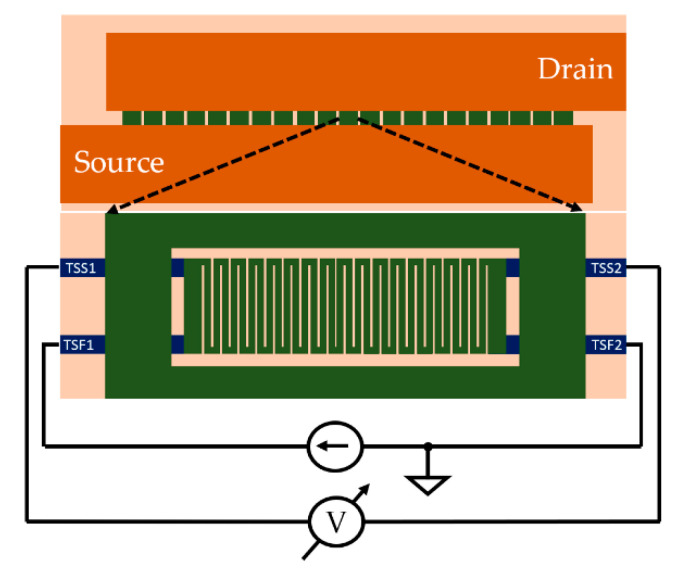
Temperature sensor layout representation.

**Figure 4 sensors-22-07223-f004:**
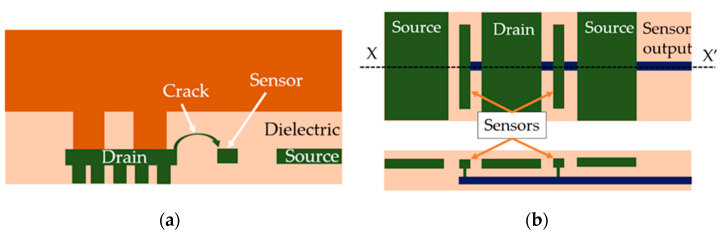
IMD crack sensor: (**a**) the working principle; (**b**) top view of the comb-like structure and cross-section along the XX’ axis.

**Figure 5 sensors-22-07223-f005:**
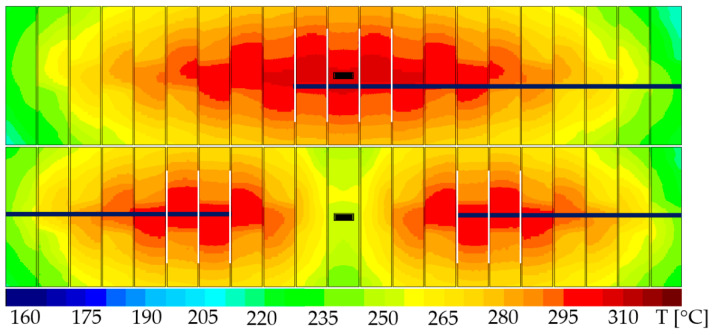
IMD cracking sensor grouping with regards to temperature distribution in TTML top—100%AA, bottom—90%AA.

**Figure 6 sensors-22-07223-f006:**
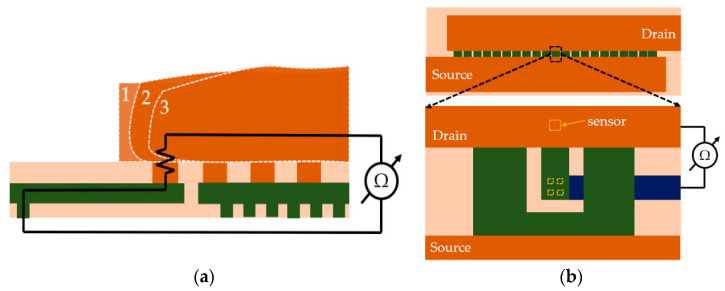
Delamination sensor: (**a**) working principle; (**b**) top view of the position on the chip and layout detail.

**Figure 7 sensors-22-07223-f007:**
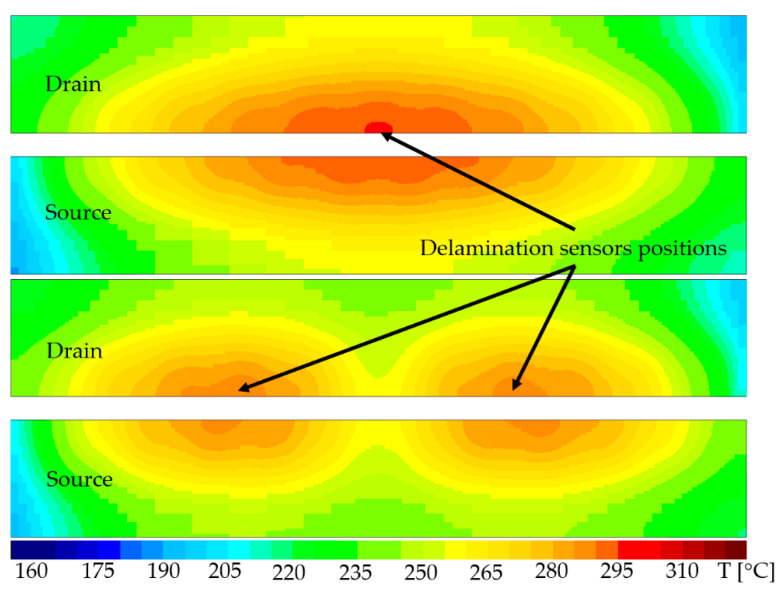
Delamination sensor placement with regards to peak temperature in PM top—100%AA, bottom—90%AA.

**Figure 8 sensors-22-07223-f008:**
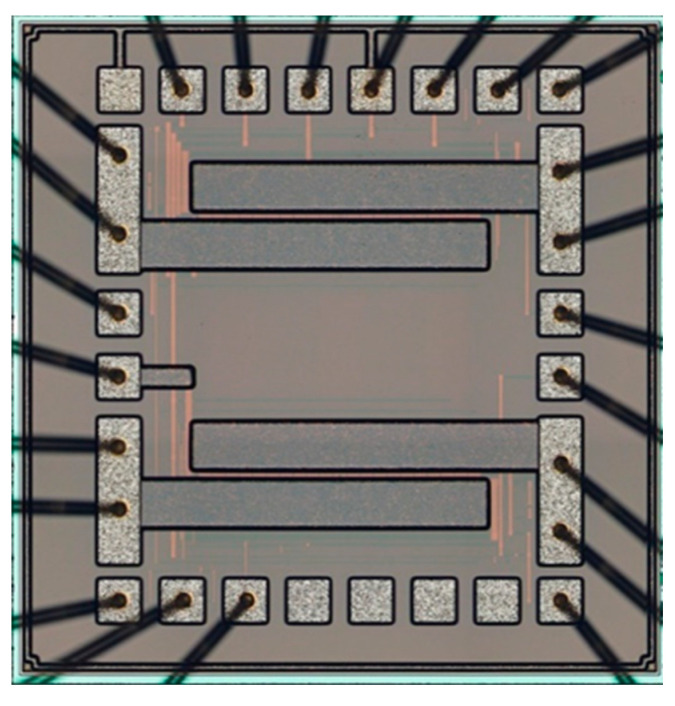
Picture of the test chip containing both structures: top—calibration structure, bottom—test structure.

**Figure 9 sensors-22-07223-f009:**
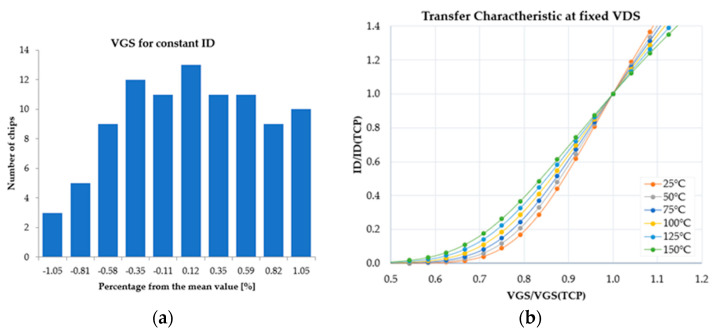
(**a**) Gate-to-source voltage spread for constant I_D_; (**b**) Normed transfer characteristic of the DMOS transistor.

**Figure 10 sensors-22-07223-f010:**
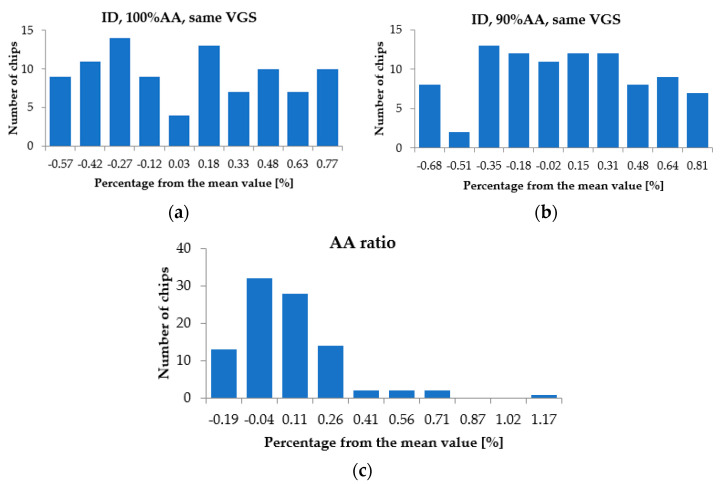
I_D_ spread for constant V_GS_ and V_DS_ for: (**a**) 100%AA; (**b**) 90%AA; (**c**) Current ratio.

**Figure 11 sensors-22-07223-f011:**
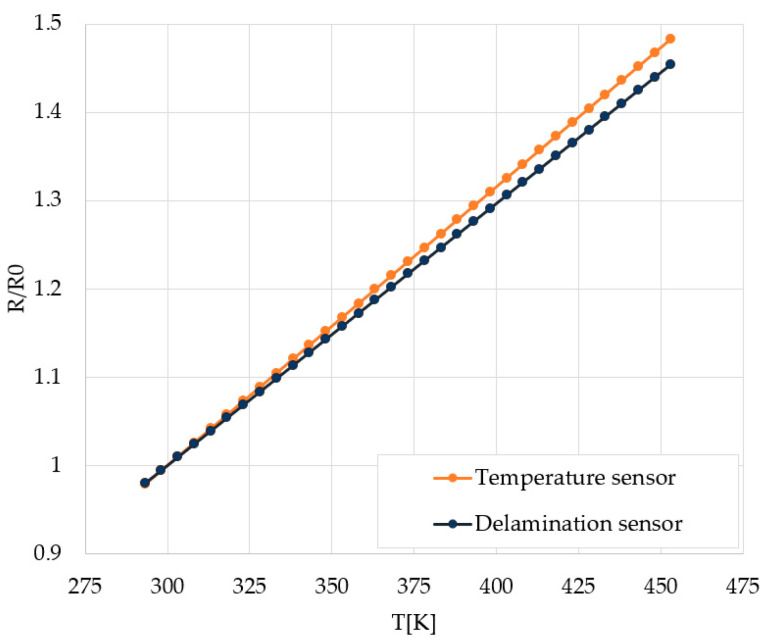
Normed temperature characteristics of temperature and delamination sensors.

**Figure 12 sensors-22-07223-f012:**
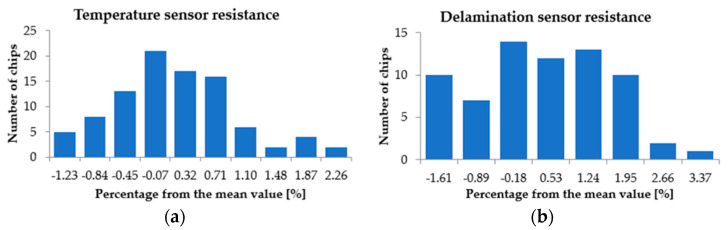
R_0_ spread for: (**a**) The temperature sensor; (**b**) The delamination sensor.

**Figure 13 sensors-22-07223-f013:**
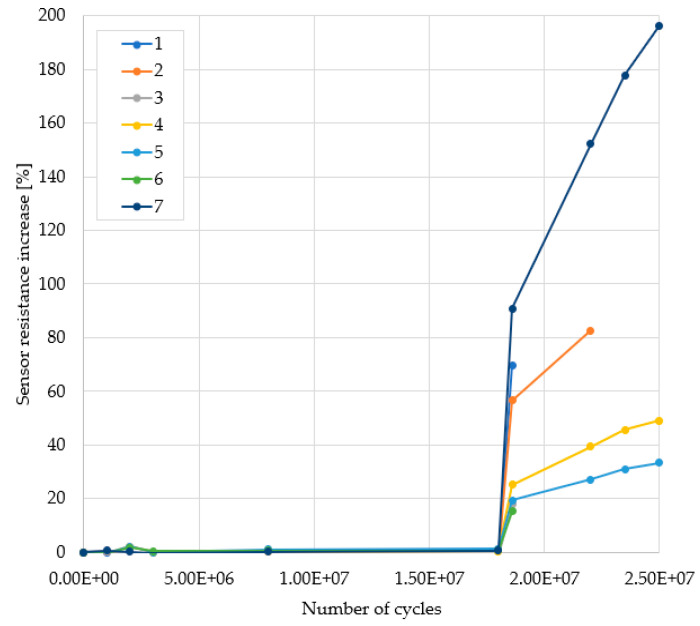
Sensor response during initial tests.

**Figure 14 sensors-22-07223-f014:**
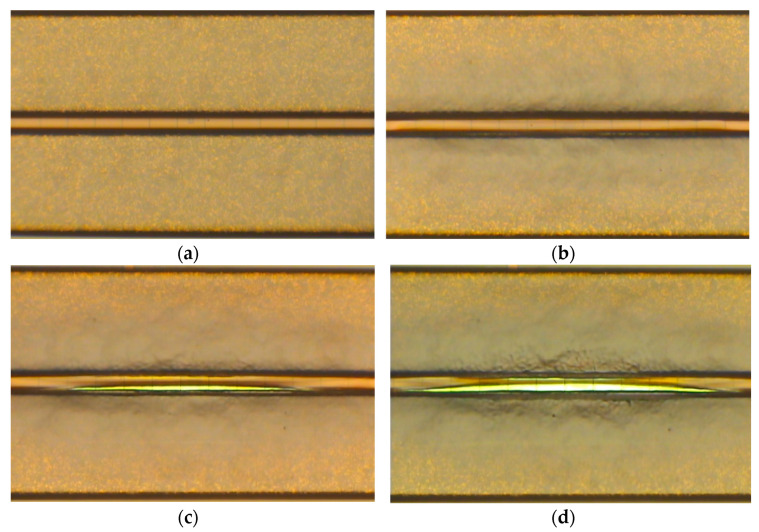
PM plates of a tested sample: (**a**) before testing; (**b**) after 10 million cycles; (**c**) after 20 million cycles; (**d**) after 30 million cycles.

**Figure 15 sensors-22-07223-f015:**
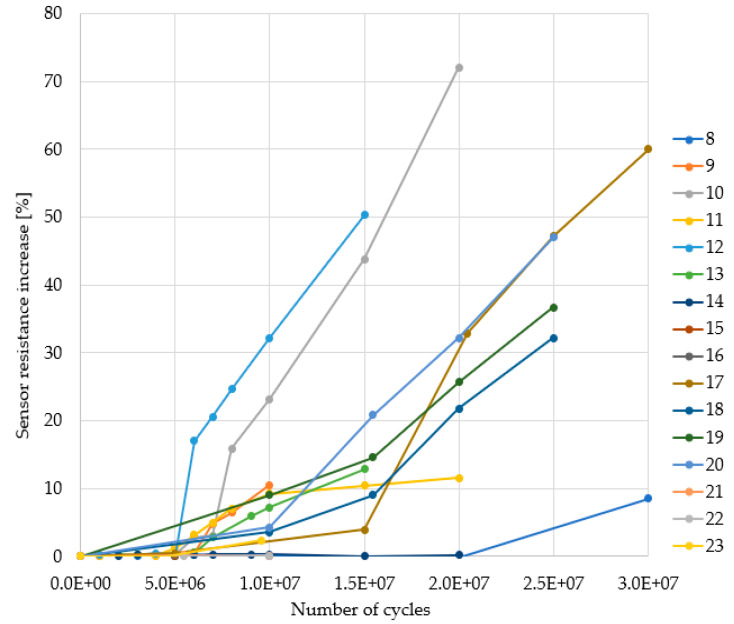
Sensor response for chips tested with 100%AA.

**Figure 16 sensors-22-07223-f016:**
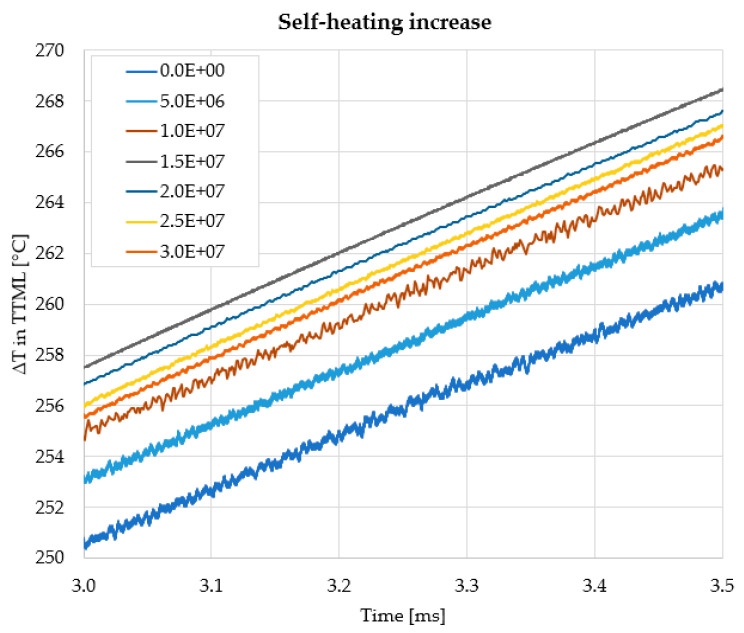
The temperature increase with respect to ambient for chips tested in the 100%AA configuration—each trace is a capture taken after 5 million cycles, as the legend suggests.

**Figure 17 sensors-22-07223-f017:**
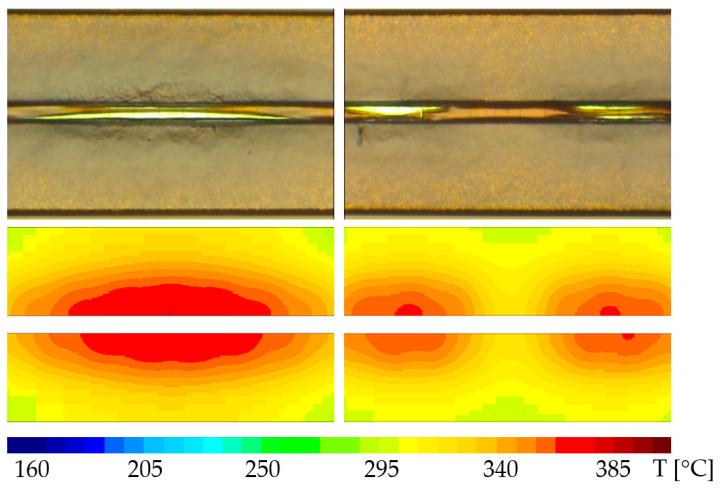
Top: Optical microscopy images for samples operating at 100%AA (**left**) and 90%AA (**right**). Bottom: Simulated temperature distribution in PM plates.

**Figure 18 sensors-22-07223-f018:**
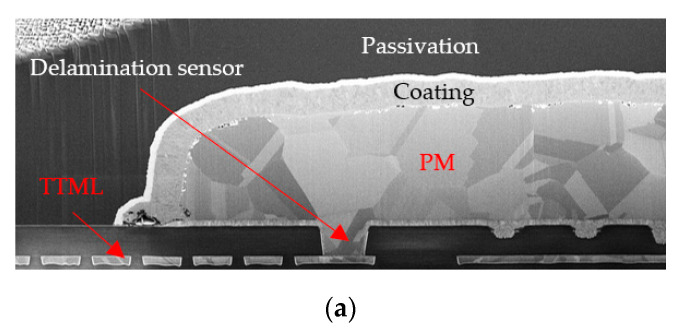
Cross-sections through the device metallization: (**a**) Unstressed sample; (**b**) Sample operated at 100%AA; (**c**) Sample operated at 90%AA.

**Figure 19 sensors-22-07223-f019:**
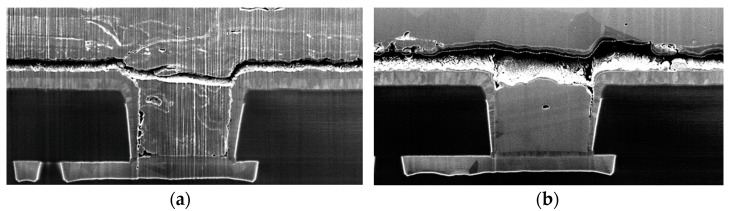
Close-up of the cross-sections of the stressed samples in the vicinity of the delamination sensor: (**a**) 100%AA; (**b**) 90%AA.

**Figure 20 sensors-22-07223-f020:**
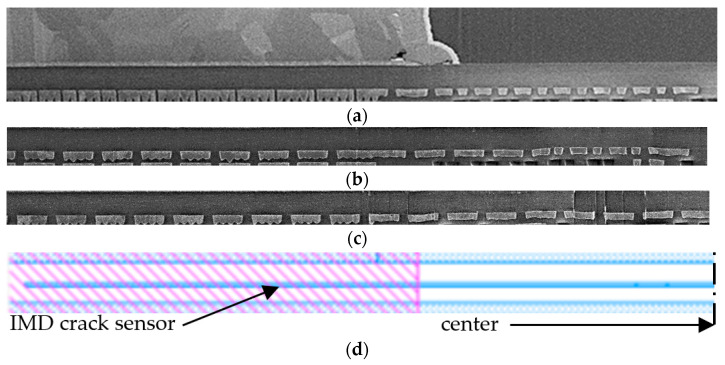
Cross-section along TTML lines: (**a**) unstressed sample; (**b**) sample cycled in 100%AA configuration; (**c**) sample cycled in 90%AA configuration; (**d**) layout.

**Table 1 sensors-22-07223-t001:** Testing conditions for trial.

	Test 1	Test 2	Test 3
T_oven_ [°C]	70	100	90
Power [W]	15	20	18
Pulse length [ms]	3.5	3.5	3.5
Pulse period [ms]	48	48	48
T_steady-state_ [°C]	125	155	150
T_sensor_ [°C]	315	465	410
Number of cycles	18 × 10^6^	5 × 10^5^	6.5 × 10^6^

## Data Availability

Not applicable.

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
