# Peer review of "Test Structure Design for Defect Detection during Active Thermal Cycling"

_sensors, 2022, doi:10.3390/s22197223_

Round 1
Reviewer 1 Report
The paper proposes a design of a power IC structure for the BCD technology aiming for metallization reliability assessment. The design integrates a temperature sensor and a mechanical sensor on the chip, so that the fatigue due to electro-thermal-mechanical phenomena leading to self-heating is assessed. Therefore, failures can be foreseen in non-critical applications and thus prevented by switching to safer operating conditions. The topic is of practical importance for monitoring defects (such as inter-metal-dielectric cracking and power metal delamination) in power ICs operating under active thermal cycling when high power transient regimes happen in very short pulses (milliseconds). The design idea is inspired from the literature or from simulations the authors published in several other papers. The testing procedure is explained, and some experimental results and failure analysis results are discussed. Finally, practical recommendations are drawn.
The paper is useful for people in industries that use power ICs. It is also well written, being very informative and of interest for other readers not directly involved in this topic.
Here there are suggestions for improvement.
- A lot of citations are only numbers, without square brackets, and this should be corrected.
- Line 162 – there is “On the other hand”, but “on the one hand” is missing.
- Line 173 – a citation is needed, where the reader can find some electro thermal simulations giving the junction temperature
- Lines 220 and 221 – please give more details about the “was chosen in such a manner” – how? and “reasonable amount of area” – what does this mean?
- Line 239, 240 – can you give details on how the temperature monitoring is done and how the decisions are made?
- Line 244 – what does “adequate placement of the delamination sensors” mean? Give some indication or references to the electro-thermal simulations you mention
- Line 307 – explain how you compute the spread (e.g. for Fig. 9b)
- In Fig 11 – which R_0 did you use? (consider that R_0 has also a spread as we can see in Fig 12)
- Fig 16 – explain in the caption what the legend means.
- Fig 18 – if possible, add some annotations on the figure so that it can be more easily followed when reading the text
Author Response
Dear madam, dear sir
Thank you for the time allocated to review our paper. All your comments were of great benefit and hopefully conducted to an improved paper. Herewith below you can find point-by-point answer to each question. For any other question or remark please do not hesitate to contact us.
Yours sincerely,
Q/A Reviewer #1
Q1. A lot of citations are only numbers, without square brackets, and this should be corrected.
Answer R#1, Q1: The problem exists in only the automatically converted pdf file. If someone downloads the word file everything is in order. We are afraid we do not know where this issue comes from. We count on the editors help to fix this issue in the final version of the paper.
Q2. Line 162 – there is “On the other hand”, but “on the one hand” is missing.
Answer R#1, Q2: Replaced “On the other hand” with “in addition”.
Q3. Line 173 – a citation is needed, where the reader can find some electro thermal simulations giving the junction temperature
Answer R#1, Q3: The references [13], [25], [26], [28] have been included in the text.
Q4. Lines 220 and 221 – please give more details about the “was chosen in such a manner” – how? and “reasonable amount of area” – what does this mean?
Answer R#1, Q4:
Suggested modification: “The temperature distribution is plotted for both cases (100%AA and 90%AA) in Figure 5. Superimposed on the temperature distribution are the sensor groups (see the thin white lines). The length and the grouping of the sensors was chosen in such a manner so that each group would contain a reasonable amount of area surrounding the high-risk areas around each hotspot (temperature that exceed 280C).”
Q5. Line 239, 240 – can you give details on how the temperature monitoring is done and how the decisions are made?
Answer R#1, Q5:
Such detailed explanations are of no importance in this section which is dedicated to the description/design of the structure and the concept of detection. Such explanations are given in Sections 3 and 4.
How the temperature is monitored?
The sensor is biased in a Kelvin connection as described in lines 167-169 and Fig 3. A constant current of 1mA is forced through the force terminals of the resistance and the voltage between the sense terminals is constantly monitored with a differential probe on the oscilloscope. The sensor is always on, given that its bias current is small, therefore no sensor self-heating occurs. This means that the resistance variation is given only by the self-heating of the DMOS transistor. Provided that the sensor characteristic is linear (Figure 11), the wave on the oscilloscope is directly proportional to the temperature. At a given number of cycles a waveform is captured (see Fig 16).
How decisions are made?
In the beginning it is unknown the degree to which the output of the PM delamination sensor and that of the temperature sensor correlate. So each were measured independently and the result was explained in section 4, lines 427 – 431
“The self-heating tends to saturate after 15 million cycles, however. Also, 9 out of 16 samples exhibit a 10% increase in the resistance of the delamination sensor before or starting with 15 million cycles. This result suggests that one could adopt as a possible failing criterion either an increase in temperature by 10°C or an increase in resistance by 10%. After this threshold is exceeded, the device ought to be switched to 90%AA to prolong safe operation.”
Q6. Line 244 – what does “adequate placement of the delamination sensors” mean? Give some indication or references to the electro-thermal simulations you mention
Answer R#1, Q6: Adequate placement means the most suitable position across the device area. The explanation is right in the following sentences and indicated in Figure 7 which is a plot resulting from an electro-thermal simulation.
Q7. Line 307 – explain how you compute the spread (e.g. for Fig. 9b)
Answer R#1, Q7: In Fig 9b is represented the transfer characteristic, no spread there. Probably you intended to refer to Fig. 9a, where the spread is calculated as: (x-μ)x100/μ, a deviation from the mean value expressed as a percentage of the mean value.
Q8. In Fig 11 – which R_0 did you use? (consider that R_0 has also a spread as we can see in Fig 12)
Answer R#1, Q8:
In Fig. 11 is represented the characteristic of one sensor. The R_0 is measured for that particular sensor. The point of spread measurements is to prove that in the end we don’t have to bother with these variations if they are small enough.
For example, in our case R0 for all chips varies between [R0_mean - 1.25%, R0_mean + 2.25%]. At high temperatures, 400°C let’s say, this would lead to a deviation from the average temperature in an interval [-8°C, +15°C], which barely makes a difference at such high temperatures.
Q9. Fig 16 – explain in the caption what the legend means.
Answer R#1, Q9:
The legend means the number of thermal cycles to which the device was subjected to previous to the moment of the capture. Explanation was added in the legend: “- each trace is a capture taken after 5 million cycles, as the legend suggests.”
Q10. Fig 18 – if possible, add some annotations on the figure so that it can be more easily followed when reading the text
Answer R#1, Q10: The picture was updated with annotations, see the red text.
Reviewer 2 Report
This paper presents a defect detection circuits for active thermal cycling, it is technically sound, but there are still some minor concerns.
1. The defects can be caused by different kind of reasons. It is necessary to rule out other scenarios inducing defects.
2. What is the number of samples of test chip? How to determine the range of temperature?(25-150 degree). It seems the trace of curves are not completed illustrated (over 1.4).
3. Please show the measurement setup and facilities.
Author Response
Dear madam, dear sir
Thank you for the time allocated to review our paper. All your comments were of great benefit and hopefully conducted to an improved paper. Herewith below you can find point-by-point answer to each question. For any other question or remark please do not hesitate to contact us.
Yours sincerely,
Q/A Reviewer #2
Q1. The defects can be caused by different kind of reasons. It is necessary to rule out other scenarios inducing defects.
Answer R#2, Q1: The paper focuses on the detection of defects which appear as a consequence of two of the most frequent failure mechanisms caused by active thermal cycling. The scope of the work is very specific. The mentioned defects are not caused by different kind of reasons other than those related to thermal cycling. Other scenarios leading to other defects are out of the scope of this work.
Q2. What is the number of samples of test chip? How to determine the range of temperature? (25-150 degree). It seems the trace of curves are not completed illustrated (over 1.4).
Answer R#2, Q2:
What is the number of samples of test chip?
The number of samples used for statistics was approximately 100. (That can be easily deducted by summing up the number of chips according to each bin in the histogram in 9a) The number of samples used in trial was 8. The number of samples subjected to thermal cycling was 16 (see Fig 15.).
How to determine the range of temperature? (25-150 degree):
We are afraid we do not really get the question. If you mean the range of temperature form sensor characterization, the answer is given in Fig.11.
If you mean the range of temperature during a single heating-cooling cycle, the answer is given in Fig. 16
Why is the interval (25-150 degree) relevant?
All information regarding relevant temperatures is summarized in Table 1.
Q3. Please show the measurement setup and facilities.
Answer R#2, Q3: The measurement setup is described in detail in reference [10]. Images of the measurement stand unfortunately cannot be provided because the company does not allow taking photographs inside their laboratory and facilities. In addition, we would consider not very helpful to the subject in the matter. The testing conditions, the device area and the packaging option should suffice for someone interested in replicating the experiment to a certain degree.